# Insights into the Evolution of *P. aeruginosa* Antimicrobial Resistance in a Patient Undergoing Intensive Therapy

**DOI:** 10.3390/antibiotics12030483

**Published:** 2023-02-28

**Authors:** Kwee Chin Liew, Jessica O’Keeffe, Heera Rajandas, Yin Peng Lee, Owen Harris, Sivachandran Parimannan, Larry Croft, Eugene Athan

**Affiliations:** 1Australian Clinical Labs, Department of Microbiology, Geelong, VIC 3220, Australia; 2Barwon Health, University Hospital Geelong, Geelong, VIC 3220, Australia; 3Centre of Excellence for Omics-Driven Computational Biodiscovery (COMBio), Faculty of Applied Sciences, AIMST University, Bedong 08100, Kedah, Malaysia; 4Deakin Genomic Centre, Deakin University, Geelong, VIC 3216, Australia; 5School of Life and Environmental Sciences, Deakin University, Geelong, VIC 3216, Australia; 6School of Medicine, Deakin University, Geelong, VIC 3220, Australia; 7Geelong Centre for Emerging Infectious Diseases, Geelong, VIC 3220, Australia

**Keywords:** WGS, antimicrobial resistance, immune evasion, SNP fitness landscape

## Abstract

Whole genome sequencing (WGS) provides insights into the evolution of antimicrobial resistance, an urgent global health threat. Using WGS, we observe evolutionary adaptation of a *Pseudomonas aeruginosa* strain within an immunocompromised patient undergoing antibiotic therapy. Two blood isolates (EA-86 and EA-87) from the patient evolved separate adaptations for antibiotic resistance, while sharing common adaptive mutations for host immune evasion. In EA-86, a silencing mutation in the antibiotic efflux pump repressor, NfxB, increased antibiotic resistance, while in EA-87, a similar mutation was seen in the antibiotic efflux pump repressor mexR. The number of genomic variants between the two isolates give a divergence time estimate of the order of 1000 generations. This time is sufficient for a bacterial lineage to have evolved an SNP in every position in the genome and been fixed if advantageous. This demonstrates the evolutionary adaptive power accessible to bacteria and the timescale for a brute-force functional survey of the SNP fitness landscape.

## 1. Introduction

Antimicrobial resistance (AMR) is a major problem worldwide and has been predicted to become one of the biggest threats to public health by the year 2050, contributing to 10 million deaths per year according to the World Health Organization (WHO) [1]. The health efforts by the global community are severely undermined by the rise in AMR and its spread in recent years. Among the more critical issues of concern is a rise in the prevalence of multi-drug resistance (MDR) in bacteria [2]. If not detected, MDR pathogens can spread quickly and lead to a higher degree of mortality [3]. This therefore necessitates a good understanding of resistance mechanisms, effective surveillance techniques and sustainable interventions [4,5]. WGS can be used as a rapid, cost-effective tool to supplement phenotypic data by providing high-resolution genomic data, instrumental in capturing genomic variations that explain the AMR mechanism(s) the isolates have used to gain resistance and evade host immune surveillance [6].

*Pseudomonas aeruginosa* (*P. aeruginosa*) is an aerobic Gram-negative bacterium that causes both community-acquired and nosocomial infections. In the intensive care unit (ICU), *P. aeruginosa* is a common cause of nosocomial infections, in 13.2–22.6% of patients [7]. For immunocompromised patients, as well as those with bronchiectasis and cystic fibrosis, infections caused by *P. aeruginosa* have been linked to high morbidity and mortality [8]. The adaptability and flexibility of the pathogen are products of the large number of virulence factors it has at its disposal, providing *P. aeruginosa* with the facility to tailor its response against different stressors in the environment. The large *P. aeruginosa* genome (~5 MB to ~7 MB), which consists of many genetic regulatory pathways, is also crucial to understanding the pathoadaptability of this pathogen, especially with current genomic techniques which allow the assessment of differences and similarities across *P. aeruginosa* populations [9]. 

Our case report, an example of how WGS complements phenotypic antimicrobial susceptibility testing (AST), sheds light on the evolutionary divergence of a *P. aeruginosa* strain obtained from an immunocompromised patient undergoing antibiotic therapy. We uncover the AMR mechanisms adopted by two isolates across the course of 18 days in a patient undergoing antibiotic therapy. This case report is in line with the WHO’s Global Antimicrobial Resistance and Use Surveillance System (GLASS), a central program focused on AMR surveillance using phenotypic data, and when available, WGS data, for several priority pathogens (including *E. coli*, *K. pneumoniae*, *S. aureus*, etc.) and will soon be extended to include *P. aeruginosa* [5].

## 2. Case Report

We present the case of Mr. BM, a 51-year-old immunosuppressed man who died from *Pseudomonas aeruginosa* sepsis and Fournier’s gangrene. Two blood isolates, EA-86 and EA-87, were obtained from the patient after 18 days of anti-pseudomonal treatment and sequenced to elucidate the source of their differing antibiotic resistance profiles.

Mr. BM was diagnosed with severe aplastic anaemia and initially responded to a first cycle of anti-thymocyte globulin, methylprednisolone and cyclosporine. The case was then complicated by haemophagocytic lymphohistiocytosis secondary to a large T-cell lymphoma. He had a background of hypertension and ulcerative colitis diagnosed in 2009, which was well-controlled with azathioprine, mesalazine and allopurinol.

During this illness, Mr. BM developed right groin lymphadenopathy and left groin ulceration. A superficial swab of left groin ulceration revealed *P. aeruginosa*, sensitive to ceftazidime, ciprofloxacin, and gentamicin; he was commenced on oral ciprofloxacin and topical clotrimazole for five days. The wound continued to progress, with increasing pain and worsening ulceration, and a diagnosis of Fournier’s gangrene was made clinically and confirmed during surgical exploration. On the day of surgical exploration, his blood cultures flagged positive for *P. aeruginosa* in three of six bottles tested. Each of the three bottles that flagged positive had two different morphologies and antibiograms *P. aeruginosa* isolates, labelled as EA-87 and EA-86.

Over the following 12 days, he underwent six operations for washout and debridement of necrotic tissue. Initial intra-operative samples of the right thigh and scrotum returned growth of *P. aeruginosa* resistant to ceftazidime and cefepime, sensitive to gentamicin and ciprofloxacin. Subsequent intrao-perative samples of the dartos muscle following further debridement revealed persistent *P. aeruginosa* resistant to ceftazidime and cefepime, sensitive to gentamicin and ciprofloxacin, and vancomycin-resistant *Enterococcus faecium* resistant to vancomycin, sensitive to linezolid. The vancomycin-resistant *Enterococcus faecium* was not covered as the dead tissue was debrided and it was not isolated in subsequent tissues and blood culture. The patient was continued on vancomycin 1 g BD and meropenem 1 g TDS to cover for other organisms given he had febrile neutropenia and Fournier’s gangrene; he was then changed to piperacillin/tazobactam 4.5 g Q8H when adequate source control was achieved. Anidulafungin 100 mg daily was used to cover for *Candida glabrata* candidemia at day 40.

He continued to receive etoposide for HLH, as well as granulocyte transfusions to support his neutrophil count during severe active infection. Repeat BMAT showed a hypocellular marrow, with extensive macrophage infiltrate, consistent with refractory HLH. Given the overall poor prognosis and the extent of the non-healing surgical wound, the decision was made to withdraw active treatment; the patient was palliated and later passed away.

## 3. Results

Different antibiotic sensitivities were found between EA-87, which was *P. aeruginosa* sensitive to ceftazidime, gentamicin and ciprofloxacin, and EA-86, which was *P. aeruginosa* sensitive to gentamicin and ciprofloxacin, but intermediate to ceftazidime and cefepime. Antimicrobial susceptibility results are shown in Table 1.

The genome size for EA-86 was found to be 6,266,206 bp, whereas that of EA-87 was 6,268,050 bp. Genome sizes differed slightly with approximately an extra 1.8 kb in EA-87 due to transposon activity (gap sizes in the assembly are small but uncertain). Only 140 genome variations were identified separating EA-86 (78) and EA-87 (62), showing recent divergence in the patient host from a common ancestral strain. Of these variations, 22 were nonsynonymous in EA-86 (Appendix A) and 16 in EA-87 (Appendix A). These isolates are most like strain YTSEY8 (GenBank accession number: CP054581.1) isolated from a hospital in China, which was used as the reference genome for variant calling. Both isolates had significant functional changes to antibiotic efflux pump repressors. EA-87 had a frame-disrupting deletion to mexR, and EA-86 had a conserved region SNP disruption to NfxB. Mutations in both genes are known antibiotic resistance-enhancing mutants [10]. Mutations in large extracellular proteins enhance immune avoidance by removing or changing epitopes. Examples are type IV pilin mutations in both isolates pilC (twice in EA-86 and once in EA-87), pilQ (EA-87) [11], a hemagglutinin-like adhesion gene (HUF04_26180, EA-87), and tpsA1, an extracellularly transported large adhesion-like gene.

The Pyoverdine system is targeted in both isolates (pvdM in EA-86 and pvdP in EA-87), an iron chelation and uptake system, and a key virulence system for *P. aeruginosa* [12]. Both mutated genes are involved in the ferribactin maturation step. Knockdown of the iron uptake pathway leads to lower iron levels and triggers a biofilm lifestyle [13]. Pseudomonas biofilms are more antibiotic- and host-resistant [14].

HUF04_31185 mutated in EA-86, likely a damX homolog, is a cell division protein, and mutants form atypical cell morphology in *E. coli* [15]. This morphology can occur under antibiotic selection [16] and is also protective against phagocytosis [17].

oprB is a carbohydrate selective porin mutated in EA-87. Porins are unlikely to mediate antibiotic uptake in *Pseudomonas aeruginosa* [18], but lack of carbohydrate in the mutant may force a more mucoid, and hence, immunologically more stealthy lifestyle. The complete set of mutations for both EA-86 and EA-87 is provided in Appendix A, respectively.

## 4. Discussion

There are several hurdles for *P. aeruginosa* colonization in a mammalian host. Even in an immunocompromised patient, there are considerable defenses against expansion, the principal one being physical barriers such as the skin and the immune system [19,20]. The host immune system surveillance recognizes the clear, non-self-handled type IV pilin and hemagglutinin, and possibly the tpsA1 protein. All are extracellular, large, structured proteins [21]. Vaccines often use these immunogenic structures as they are, first and foremost, markers of bacterial infection for the patient’s immune system to recognize. The original infectious strain is unlikely to have been adapted to infecting humans, but has gained this functional ability by mutation of key genes while in the patient. Under the strong selective pressure of antibiotic treatment of the patient, the strain evolved antibiotic resistance in two separate lineages, represented by the isolates EA-86 and EA-87. One lineage, EA-86, evolved a mutation changing a highly conserved domain amino acid in NfxB, a repressor of the mexCD-opr, a multidrug efflux operon—a classic response to antibiotic challenge, even seen in vitro [22]. The other lineage, EA-87, introduced a frame-breaking mutation in mexR, also a repressor for the mex efflux operon. Both are known antibiotic resistance-enhancing mutations. The small number of SNPs different between EA-87 (49) and EA-86 (36), with an approximate mutation rate of one SNP per 1000 generations [23], means of the order of 10,000 generations (approximately four months worth) since divergence of the two isolates. The timescale for a brute-force (in the cryptographic sense) survey of the fitness landscape of the six million SNPs which can be varied in the *P. aeruginosa* genome is of the order of 1000 generations, as at this timescale, the population doubling at every generation has produced a mutant for every base in the genome.

## 5. Materials and Methods

### 5.1. Microbiological Testing

Blood cultures were incubated in BD BACTEC^TM^ Instrumented Blood Culture System and microorganisms from positive blood cultures were processed via standard methods on horse blood, chocolate, and MacConkey agar plates. Species identification was performed using the Bruker MALDI Biotyper with Bruker 38 Biotyper 3.4 software and library version V4.0.0.1 (Bruker Daltonik, Bremen, Germany), and antimicrobial susceptibility testing (AST) was undertaken using Vitek 2 semiautomated system (version 08.01; bioMérieux, Durham, NC, USA) with the commercially available Vitek 2 AST-N246 card.

### 5.2. DNA Extraction and Sequencing

DNA extraction of the isolates was performed using the HiYieldTM Genomic DNA Mini Kit (Real Biotech Corporation, Taiwan) and Nextera DNA Flex (Illumina, San Diego, CA, USA). Library preparations were made for each sample according to the standard protocol. The libraries were subjected to whole-genome sequencing using the NovaSeq 6000 Sequencer (Illumina, San Diego, CA, USA) according to the manufacturer’s instructions. Approximately 2 GB of 2 × 150 nt paired-end reads were generated for each sample.

### 5.3. Bioinformatics Analysis

Reads were cleaned and trimmed using fastp with default parameters. The genome assembly of cleaned reads and annotation were undertaken using Spades 3.14.1 [24] and Prokka v1.14.5 [25], respectively, using default parameters. Snippy v4.6.0 [26] was used to call SNPs based on the nearest reference genome, *P. aeruginosa* strain YTSEY8 (GenBank accession number: CP054581.1). The GenBank file for YTSEY8 was used as input to Snippy to generate annotated variant changes for each genome.

## 6. Conclusions

As WGS becomes cheaper and genomic analysis easier, it becomes possible to not only track strains, but also follow bacterial evolution directly and within patients. These adaptations, including to clinically relevant phenotypes such as antimicrobial resistance, provide a great deal of information on the mechanisms bacteria use to avoid immune surveillance, adapt to antibiotic therapy, and become more virulent. This “real-time” information can be harnessed to improve clinical management, such as by using combination therapies and “second-guessing” the evolutionary adaptation strategy that bacteria use. Bacteria evolve either by plasmid/transposon/viral rearrangements and genomic additions/deletions, or by single nucleotide mutations of the core genome. In this case report, the initial infection strain has adapted by a small number of SNP mutations. As bacterial traits are governed mostly by the genome sequence (epigenomics also plays a part), bacterial evolution can be seen as a search of a phenotype fitness landscape encoded by the approximately six million genomic bases. The three-week duration of the infection corresponds to the time required for an exhaustive, brute-force search of the fitness landscape by SNP/indel mutation of all bases in the genome (1000 generations, or about 21 days with half-hour generations). In both isolates, single SNP/indel events (in mexR and NfxB) would have significantly increased antibiotic resistance. Thus, without the extracellular introduction of antibiotic resistance plasmids, etc., the typical time to develop antibiotic resistance will be about three weeks at 37 °C. To evolve proteins with new functions (approximately 50% of bases changed in a CDS) requires a chain of many sequential SNP changes. As the average gene is 3 kb long, 1500 SNPs would be required sequentially—of the order of one million generations, or about 60 years. Plasmid- and viral-mediated uptake of novel functional peptides will be the principal mode of antibiotic adaptation during transmission from patient to patient. Intra-patient, where plasmid- and viral-mediated DNA transfer is limited, the principal mode of adaptation will be SNPs and indels, with adaptation time by evolution of novel protein functions measured in decades.

## Figures and Tables

**Table 1 antibiotics-12-00483-t001:** Antimicrobial susceptibility of EA-86 and EA-87 blood culture isolates using VITEK 2 systems version 08.01 and AST-N246 card. (I—intermediate; R—resistant; S—susceptible.)

Antibiotics	EA-86 MIC (mg/L)	EA-87 MIC (mg/L)
Timentin	R ≥ 128	R ≥ 128
Pipercillin/tazobactam	S 32	S 32
Ceftazidime	I 16	S 8
Cefepime	I 8	S 8
Meropenem	S 1	S 0.5
Amikacin	S ≤ 2	S ≤ 2
Gentamicin	S ≤ 1	S ≤ 1
Tobramycin	S ≤ 1	S ≤ 1
Ciprofloxacin	S 1	S 0.5

## Data Availability

The sequencing raw data has been submitted to the Sequence Read Archive (SRA) database in NCBI under bioproject number PRJNA906317.

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
