# Peer review of "Insights into the Evolution of P. aeruginosa Antimicrobial Resistance in a Patient Undergoing Intensive Therapy"

_antibiotics, 2023, doi:10.3390/antibiotics12030483_

Round 1

Reviewer 1 Report

This case report is well-information for bacterial evolution in an immunocompromised host. It is a good report that can be published in the Antibiotics journal with minor correction and responses as below;

1.      There are several typo errors. Some will be mentioned, but not all.  Please double check throughout the manuscript.

2.      Writing scientific names of microorganisms should be followed e.g. italics, full name at the first mention (e.g. Line 100, PAO1). Please double check throughout the manuscript.  

3.      Line 24, “A difference seen in the colony morphologies” This is not mentioned anywhere in the manuscript. Is it really meaningful?

4.      Lines 66-67, how about the AST result for cefepime at this stage? Resistant?

I think it should be mentioned here as you mentioned in Lines 75-76 that P. aeruginosa returned to resistant to this drug.

5.      It is consistent for drug name. e.g. Line 67, “..oral Ciprofloxacin and topical Clotrimazole for five days” Need Capital letter?

6.      Lines 71-72, “cultures flagged positive with three of six bottles positive with P. aeruginosa, labelled as 71 EA-87 (P. aeruginosa), EA-86 (P. aeruginosa)”

Did you recover three positive bottles of six positive bottles? Why did you get only two isolates, not three isolates? Please explain.

7.      Add one Tab at Line 83

8.      Line 93, an extra 1.8 Kb should be more exactly?

9.      “hemagluttinin gene?” or “hemagglutinin” What is gene name of hemagglutinin? Please specify and add in Line 99.

10.  Line 118 PA01 or PAO1

11.  Line 124, What is PA? P. aeruginosa? Have you mentioned this acronym before?

12.  Materials and Methods Part, please add city and country for instruments and kits, e.g. Lines 152, 157 (Like Lines 155-156), 159, 162.

13.  Line 192, 3Kb long for Gene or Proteins?

14.  Line 204, check font please

15.  Line 217, the link is not found

16.  Acknowledgements should be written

17.  Please recheck references in MDPI style

Author Response

We thank Reviewer 1 for the valuable comments. We have addressed all the comments and provided the response as an attachment.

Reviewer 2 Report

1.       A clinical case is not strictly defined and there are many ambiguities.

·       Why is vancomycin used in treatment when a vancomycin-resistant Enterococcus feacium strain has been isolated.The application of meropenem is given, while the results of susceptibility to this antibiotic are not given. Results are also missing for piperacillin/tazobactam. Why was anidulafungin administered?

·       The names of bacteria are generally written in italics.

·         The wrong name of the antibiotic, ceftazadime, is used - apparently ceftazidime is meant.

·         Why are the names of antibiotics written with a small letter at the beginning of the word and at the same time with a capital letter at the beginning of the word?

·         What is meant by van A, the name for a gene? (then italics should be used) or phenotype designation (then VanA phenotype is correct).

·         I can not find any MICs values for isolates EA-87 and EA-86.

2.    It would be advisable to add a table characterizing the isolates analysed– any virulence factors, resistance genes, ST types – when there is the SNP analysis, ST types should be listed too.

3.       Row 93 and 94: “Genome sizes differed slightly with an extra 2kb in EA-87 due to transposon activity. Annotation of the genomes found no evidence of plasmids.“ What bioinformatics tool led you to conclude that no plasmid was present? A transposon can be present in a chromosome but also in a plasmid. Also to evaluate the presence of plasmids - the NovaSeq 6000 Sequencer (Illumina) can only provide us with short reads - for the evaluation of plasmids, we need another sequencing technique such as Minion Illumine, which provides us with long reads and then based on a combination of long and short reads, evaluate with another bioinformatics tool, whether there are plasmids in the sample, or of what size. How did you proceed when analysing sequencing data to evaluate the presence of mobile genetic elements?

4.       It is not clear from the text - how many SNPs were identified between the two isolates EA-87 and EA-86, there are demonstrated only the SNPs differing from the reference genome.

5.       Figure 1 - the comparison of SNPs should be illustrated with the help of some bioinformatic software with precise identity comparison between the isolates themselves.

6.       Supplementary data - it should be attached to the manuscript.

Author Response

We thank Reviewer 2 for the valuable comments. We have addressed all the comments and provided the response as an attachment.

Round 2

Reviewer 2 Report

First of all, I thank the authors of the manuscript for editing the text based on my comments.

The text has been adequately edited and I am now pleased to recommend the manuscript for acceptance.